# Risk Factors and Clinical Characteristics of Patients with Ocular Candidiasis

**DOI:** 10.3390/jof8050497

**Published:** 2022-05-11

**Authors:** Toru Sakamoto, Kenji Gotoh, Kenyu Hashimoto, Chiyoko Tanamachi, Hiroshi Watanabe

**Affiliations:** 1Department of Infection Control and Prevention, School of Medicine, Kurume University, Kurume 830-0011, Japan; sakamoto_tohru@kurume-u.ac.jp (T.S.); hwata@kurume-u.ac.jp (H.W.); 2Department of Clinical Laboratory Medicine, Shin Koga Hospital, Kurume 830-8577, Japan; kenyu.hashimoto@gmail.com; 3Graduate School of Medicine, Kurume University, Kurume 830-0011, Japan; 4School for Medical Technology, Kurume University, Kurume 830-0011, Japan; tanamachi_chiyoko@kurume-u.ac.jp

**Keywords:** ocular candidiasis, fungal endophthalmitis, endogenous endophthalmitis, *Candida*, (1,3)-β-D-glucan (βDG), ophthalmologic examination

## Abstract

Ocular candidiasis is a critical and challenging complication of candidemia. The purpose of this study was to investigate the appropriate timing for ophthalmologic examinations, risk factors for complications of ocular lesions, and their association with mortality. This retrospective cohort study applied, using multiple logistic regression analysis and Cox regression models, to cases of candidemia (age ≥ 18 years) for patients who underwent ophthalmologic consultation. Of the 108 candidemia patients who underwent ophthalmologic examination, 27 (25%) contracted patients had ocular candidiasis, and 7 experienced the more severe condition of endophthalmitis, which included subjective ocular symptoms. In most cases, the initial ophthalmologic examination was performed within one week of the onset of candidiasis with a diagnosis of ocular candidiasis, but in three cases, the findings became apparent only after a second examination within 7–14 days after onset of candidiasis. The independent risk factor extracted for the development of ocular candidiasis was the isolation of *C. albicans* (OR, 4.85; 95% CI, 1.58–14.90)*,* unremoved CVC (OR, 10.40; 95% CI, 1.74–62.16), and a high βDG value (>108.2 pg/mL) (HR, 2.83; 95% CI = 1.24–6.27). Continuous ophthalmologic examination is recommended in cases of candidemia with the above risk factors with an initial examination within 7 days of onset and a second examination 7–14 days after onset.

## 1. Introduction

Candidemia is a nosocomial infection that is a major cause of morbidity and mortality [1,2]. More than 250,000 patients are affected worldwide every year, and more than 50,000 of them die due to this critical infectious disease [3]. A comprehensive candidemia care bundle definitely improves patient care and mortality in patients with this severe infectious disease [4,5,6,7]. Generally, a comprehensive candidemia care bundle includes the appropriate choice of antifungal drugs, the removal of central venous catheters, and ophthalmological examinations. Candidemia sometimes leads to hematogenous disseminated lesions in several parts of the body, and metastatic ocular infection is a particularly challenging complication [8]. Therefore, ophthalmological examination is strongly recommended in all the preventive guidelines of invasive candidiasis [9].

All the preventive guidelines, however, include a few detailed recommendations about when and how often ophthalmological examination should be performed. Although both the necessity and utility of repeated ophthalmological examination has previously been reported, specific guidelines are yet to be published. Furthermore, there have been few reports of the risk factors associated with ocular candidiasis. One factor is that the relationship between ocular candidiasis and prognosis in patients with candidemia is not well known.

Here, we describe a retrospective cohort study to determine the risk factors related to ocular candidiasis and identify the relationships between mortality and eye involvement. In addition, we analyze the clinical characteristics of patients with ocular involvement in order to establish appropriate intervention methods via ophthalmological examination.

## 2. Materials and Methods

### 2.1. Study Design

This retrospective cohort study was conducted from April 2013 to March 2020 at the Kurume University Hospital in Japan, which is a tertiary-care university hospital with more than 1000 beds. We used a microbiology database to identify all patients (age ≥ 18 years) with candidemia using microbiology database and collected clinical information from their medical records. This study was approved by the Medical Ethics Committee of the Kurume university hospital (No 21123) and was implemented in accordance with the Declaration of Helsinki. Patient consent was waived owing to the nature of a retrospective study.

### 2.2. Clinical Definitions

Candidemia was defined as isolation of *Candida* spp. from at least one blood culture in a patient with clinical sign of infection. Subsequent positive cultures from the same patient were considered as a new episode if an interval of more than 30 days had transpired the two episodes. Exclusion criteria included patients who were less than 18 years of age, patients considered contaminated and untreated, and patients who had received no ophthalmological consultation. The onset of candidemia was defined as the day the initial positive blood sample was drawn, which subsequently formed a culture positive for *Candida* spp. To classify ocular candidiasis, we referred to criteria established in previous studies [10,11,12,13]. Proven ocular candidiasis was defined as ocular lesions combined with a positive histology or a culture of vitreous aspirate. Probable Candida endophthalmitis was defined as vitritis or fluffy lesions with extension into the vitreous. Probable Candida chorioretinitis was defined as deep focal white infiltrates in the retina. In addition, hemorrhages, Roth spots, or nerve fiber layer infarctions (cotton wool spots) in patients with candidemia were classified as probable Candida chorioretinitis if no other causes for these abnormalities were present (e.g., diabetes mellitus or hypertension). If signs of chorioretinitis were seen in patients with underlying systemic disease that could cause similar lesions (e.g., diabetes, hypertension, or concomitant bacteremia), these cases were classified as possible ocular candidiasis. In statistical analysis, we grouped all cases as either “ocular candidiasis” or “non-ocular candidiasis” according to the above-mentioned classification. The ocular candidiasis group included patients with proven ocular candidiasis, patients with probable Candida endophthalmitis, and patients with probable Candida chorioretinitis. The non-ocular candidiasis group, on the other hand, included patients with possible ocular candidiasis and patients with no abnormal ocular findings. In previous research, a classification of possible ocular candidiasis has often included patients with ocular candidiasis. In the present study, we assigned possible ocular candidiasis to the non-ocular candidiasis group in this study to insure a strict assessment of risk factors [10].

### 2.3. Underlying Conditions and Clinical Status

The predisposing factors and clinical information acquired from medical records included age, sex, underlying diseases (diabetes mellitus, hypertension, chronic heart disease, chronic obstructive pulmonary disease, liver diseases, chronic kidney disease, malignant disease), immunocompromised status (steroid therapy, neutropenia (absolute neutrophil count <500/µL), use of immunosuppressive agents, exposure to chemotherapy, exposure to radiation therapy, HIV infection, recipients of stem cell transplantation), intensive care unit (ICU) admission, history of surgery, prior antibiotics exposure, prior antifungal exposure, presence of a central venous catheter (CVC), removal of the CVC, interval onset of candidemia and removal of CVC, total parenteral nutrition (TPN), mechanical ventilation, hemodialysis, septic shock, initial antifungal drugs, interval onset and administration of antifungal drugs, causative *Candida* spp., persistent blood stream infection (blood culture again positive after an interval of at least 72 h from onset), and a prognosis of 30-day mortality. The ophthalmological clinical course was also collected from the medical records. Ophthalmological clinical information includes the following: when was the ophthalmologic examination performed, counting from the onset of candidemia; how many ophthalmologic examinations resulted in a diagnosis of ocular candidiasis; what are the ophthalmoscopy findings; is there a presence of subjective symptoms of the eyes; and what is the ophthalmology treatment history. The (1,3)-β-D-glucan (βDG) test values were evaluated via the WAKO β-glucan test (Wako, Tokyo, Japan), and the highest value measured within 7 days of the onset of candidemia was used.

### 2.4. Statistical Analysis

Continuous variables were presented as the mean ± standard deviation (SD) or interquartile range (IQR), and these were compared using either a Student’s *t*-test or a Mann−Whitney U test. For the values of βDG, logistic single-regression analysis was used for comparison. Categorical variables were presented as numbers and percentages and compared using either the χ^2^ test or Fisher’s exact test. We compared demographic characteristics, clinical factors, and outcomes between episodes with and without ocular candidiasis by using univariate analysis, and multivariate analysis was conducted using items that were statistically significant (*p*-value < 0.05) in univariate analysis. Multivariate statistical analysis was conducted by using logistic regression multivariate analysis with odds ratios (OR) and 95% confidence intervals (CI). A more detailed statistical analysis was performed for βDG. Receiver operating characteristic (ROC) curves for βDG values were described, and their cut-off values were determined with the maximum Youden index. All cases were divided into groups with either high or low βDG values based on the cut-off values, and the relationship between βDG values and ocular candidiasis was illustrated using a Kaplan−Meier estimator. Cox regression models were used to calculate the adjusted hazard ratios (HRs) with a 95% confidence interval. The association between 30-day mortality and ocular candidiasis also was studied using the same methods for statistical analysis. Two-tailed *p*-values of <0.05 were considered statistically significant. Statistical analyses were performed using R program language, version 4.1.3, and JMP software, version 15.

## 3. Results

### 3.1. Classification of Ocular Candidiasis

The process for selecting target cases appears in Figure 1. We identified 149 cases of candidemia and excluded 11 cases; the 11 cases included 8 pediatric cases and 3 untreated cases, which were considered contamination. Of the 138 cases of candidemia collected from the medical records of adult patients (age ≥ 18 years), 30 cases with no ophthalmological examination were also excluded. Almost all cases of patients with no ophthalmological examination were severe, and those people either died before diagnosis of candidemia, died before scheduled ophthalmological examination, or withdrew from treatment due to a bad prognosis. Among the remaining 108 cases of patients who had undergone ophthalmological examination, abnormal findings were found in 40 cases (37%) via fundoscopy. In these 40 cases were included 7 cases of probable Candida endophthalmitis, 20 cases of probable Candida chorioretinitis, and 13 cases of possible ocular candidiasis according to the criteria as stated above. According to the definition mentioned above, 27 cases, which included 7 with probable Candida endophthalmitis and 20 with probable Candida chorioretinitis, were defined as ocular candidiasis in statistical analysis. Actually, the ophthalmologic findings in cases with possible ocular candidiasis in this study were more likely due to other diseases such as diabetic retinopathy rather than to fungal causes. No changes in fundus findings were observed in these patients during ongoing ophthalmologic examinations.

### 3.2. Clinical Characteristics of Ocular Candidiasis

Patient backgrounds, ocular symptoms, and timing of ophthalmologic consultations for 27 patients with ocular candidiasis appear in Table 1. Seven patients with probable Candida endophthalmitis complained of subjective eye symptoms. Six patients presented with poor eyesight, two cases presented with myodesopsia, and one case presented with misty vision. Most patients with ocular symptoms received an ophthalmologic examination and a diagnosis of vitritis within 2 days of the onset of subjective symptoms. On the other hand, some cases complained having been aware of eye symptoms for 4–5 days before screening ophthalmologic examination. These patients did not consider ocular symptoms important because they were not asked about eye symptoms. Generally, ocular candidiasis was found by ophthalmological examination after Candida was isolated from a blood culture, but candidemia was suspected in two of the cases with subjective eye symptoms based on fundus findings before *Candida* spp. was isolated. When one of these two cases had an episode of fever, their blood culture was not submitted, and a central venous catheter was removed, which could have led to a delay in the diagnosis of candidemia. Of the seven patients with probable Candida endophthalmitis, all presented with vitreous opacity, and of them, six patients had bilateral lesions. No cases required surgical treatment of the eye, but one case was treated by direct administration of Voriconazole (VRCZ) into the vitreous. Twenty patients with probable Candida chorioretinitis did not complain of subjective eye symptoms. Fundus examination identified 18 patients with abnormal bilateral findings, and 2 patients had abnormal findings only on one side. All cases presented with cotton wool spots, and some showed petechial hemorrhage.

Primary ophthalmological examinations were conducted an average of 5.6 ± 3.7 days following onset of candidemia, and the median was 5 days. A second examination was performed an average of 10.1 ± 4.8 days following the first, and the median was 8.5 days. Of the 27 patients, 6 died before the second ophthalmologic examination, and the remaining 21 patients all had two or more repeated ophthalmologic examinations. Most cases showed improvement in fundus findings in the second and in subsequent ophthalmologic examinations, but three cases showed a tendency toward exacerbation. In these three cases, second ophthalmological examinations were conducted in 8, 9, and 12 days, respectively, after candidemia had occurred. It is important to emphasize that in these three cases, no abnormalities in the ocular examination were evident during the first examination, but the abnormal findings became apparent during the second examination. In particular, in the patient with AML and severe neutropenia, the ocular lesions gradually worsened with each ophthalmologic examination despite the administration of appropriate antifungal medications, which eventually led to treatment with intravitreal injections.

Three of all patients showed no abnormal findings during the first ophthalmologic examination but were diagnosed with ocular candidiasis during the second examination. One of these three cases presented with severe neutropenia and was eventually treated by vitreous injection due to a gradual exacerbation of eye symptoms.

### 3.3. Analysis of Risk Factors for Ocular Candidiasis

The characteristics of the patients appear in Table 2 First, we composed an overall picture of the 108 cases analyzed in this study. The average age was 67.8 ± 12.5, and the percentage of men was 67%. As for the choice of inpatient wards, 38.8% of the patients were admitted to the intensive care unit (ICU). In terms of underlying disease, malignant tumors accounted for the largest percentage of patients (51.8%), with diabetes mellitus second (33.3%). As for immunosuppressive factors, 29.6% were receiving chemotherapy, and 17.5% were receiving steroid therapy. Almost one-quarter (24%) of the patients had undergone laparotomy or open-heart surgery within the past month. The breakdown of fungal species was as follows: *Candida albicans* was the most commonly isolated species (42.6%), followed by *Candida parapsilosis* (25.0%), *Candida glabrata* (14.8%), *C. famata* (5.6%), *C. tropicalis* (4.6%), and *C. krusei* (4.6%). Of the strains detected in this analysis, none with known drug susceptibility were suspected to be highly resistant to antifungal drugs such as fluconazole or echinocandin. Initial antifungal drugs were as follows: azole (59.3%), echinocandin (37.0%), and liposomal amphotericin B (2.8%). Persistent blood stream infection accounted for 36.2%. Among the cases with persistent candidiasis, there was one case each with suspected complications of infectious endocarditis and vertebral osteomyelitis. The rate for 30-day mortality was 20.4%.

In univariate analysis, unremovable CVC, and isolation of *C. albicans*, βDG value were statistically significant, and multivariate analysis was performed according to the method defined. The results indicated that isolation of *C. albicans* (OR, 4.85; 95% CI, 1.58–14.90)*,* unremoved CVC (OR, 10.40; 95% CI, 1.74–62.16), and a high βDG value (OR, 1.003; 95% CI, 1.0004–1.005) were detected as independent risk factors for ocular candidiasis (Table 3). With respect to the types of catheters that were not removed, the percentage of central venous access ports (CV port) was high (50%).

Multivariate analysis was performed using items that ware statistically significant (*p* value < 0.05) in the univariate analysis. All items were statistically significant (*p*-value < 0.05) also in multivariate analysis.

### 3.4. βDG Values and Ocular Candidiasis

As mentioned above, we determined that a high βDG level is an independent risk factor in the development of ocular candidiasis. Therefore, we further analyzed the relationships between βDG levels and ocular candidiasis. Receiver operating characteristic (ROC) curves for βDG values were developed, and their cut-off values were determined using the maximum Youden index of 108.2. The area under curve (AUC) was calculated to be 0.68. We divided the patients into two groups: βDG > 108.2 pg/mL and <108.2 pg/mL. The distribution of time to diagnose ocular candidiasis by ophthalmological examination was estimated using the Kaplan−Meier estimator and analyzed by log-rank testing (Figure 2). The log-rank test showed *p* < 0.05, which was statistically significant. We used Cox regression models to calculate the adjusted hazard ratios (HRs) and 95% confidence intervals. High βDG values (>108.2 pg/mL) were statistically significant for the risk factors of ocular candidiasis (HR = 2.83; 95% CI = 1.24–6.27).

### 3.5. 30-Day Mortality and Ocular Candidiasis

We also investigated the association between complications of ocular candidiasis and 30-day mortality. Patients were divided according to their 30-day mortality rates and for a comparison of backgrounds, which included complications from ocular candidiasis (Table 4). In univariate analysis, female, unremoved CVC, persistent BSI, ocular candidiasis, and βDG values were statistically significant, and in multivariate analysis, unremoved CVC (OR, 17.76; 95% CI, 2.18–144.38) was detected as an independent risk factor for 30-day mortality (Table 5). Though ocular candidiasis was not an independent risk factor in 30-day mortality, the mortality rate for patients with ocular candidiasis was 37.0% compared with 14.8% for the patients without ocular candidiasis.

Multivariate analysis was performed using items that ware statistically significant (*p* value < 0.05) in the univariate analysis. Unremoved CVC was statistically significant (*p* value < 0.05) in multivariate analysis.

## 4. Discussion

Ocular candidiasis is a critical complication of candidemia. Because ocular candidiasis can lead to blindness, close attention should be paid to the complications it causes in candidemia and ophthalmologic evaluation is strongly recommended [14]. The purpose of this study is to propose a more effective and comprehensive treatment strategy for candidemia by identifying the risk factors for ocular candidiasis in patients with candidemia and how those risk factors correlate with prognosis as well as determining the appropriate time for an ophthalmologist to perform a fundus examination. For most cases in the present study, ocular candidiasis was diagnosed within 7 days after the onset of candidemia. In some cases, however, the fundus findings became apparent 7–14 days after onset, which essentially closed many of the so-called windows of opportunity. Furthermore, in the present study, independent risk factors for ocular candidiasis were reported as follows: isolation of *C. albicans*, unremovable catheters after the onset of candidiasis, and a high βDG value. Although not statistically significant, mortality tended to be higher in cases complicated by candida eye lesions.

The incidence of ocular candidiasis has been significantly reduced over the past few decades due to promotions for the use of appropriate antifungal drugs [15]. Nonetheless, previous studies have reported incidences of ocular candidiasis that range from 2 to 46% [10,12,16,17]. In the present study, the complication rate of ocular candidiasis was 25%, which did not differ significantly from previous reports. We presented patients with candidemia who had complained of ocular symptoms and who had complications from advanced vitritis, which suggests that ocular subjective symptoms should always be checked in cases of candidemia. Here, we must reiterate that the basic policy is to perform ophthalmologic examinations because there are cases in which patients do not complain of ocular symptoms, such as those who are admitted to the ICU. Candida eye lesions can be divided into either vitritis or retinitis depending on the depth of the lesion. The condition commonly referred to as endophthalmitis corresponds to vitritis. It is important for a clinician to distinguish between these two types of inflammation, as they may lead to different treatment strategies. Vitritis is sometimes treated surgically or by intravitreous injection of antifungal drugs. In the present study, only one case was treated by intravitreous injection [14]. With respect to the selection of antifungal agents, it is necessary to select options that have shown translocation into the vitreous. Echinocandin is not recommended for use in patients with ocular lesions due to poor vitreous migration although there are reports that it can be expected to migrate into the chorioretina [18,19,20]. Early ophthalmologic consultation within 7 days of onset is recommended, and some reports indicate that diagnoses have been made after the 7th day [9,10]. In the present study, 23 cases were diagnosed within 7 days of onset, while 3 cases were diagnosed during the second examination that occurred 7–14 days following diagnosis when there were no findings during the first examination. Of the three cases, the case with the most slowly developing ocular lesions was characterized by severe neutropenia. Some reports have recommended that patients with neutropenia undergo ophthalmologic examination after their neutrophil counts have recovered [21,22]. Repeat fundus examinations are more important in neutropenic patients because these cases are less likely to have an immune response, which could delay the manifestation of ocular lesions.

We have shown that isolation of *C. albicans* is an independent risk factor for ocular candidiasis. This result is similar to previous reports [12,22,23]. Abe et al. reported a strong association between *C. albicans* and ocular candidiasis due to greater capacity for invasion, induction of inflammatory mediators, and the recruitment of both neutrophils and inflammatory monocytes [24]. In addition, *C. albicans* also accounted for a major proportion of all cases in this study. Although there are some reports of an increase in non-albicans candida, *C. albicans* remain to be the most frequently detected organism in our facility [25]. It has been reported that *C. parapsilosis* and *C. glabrata* are less likely to cause ocular candidiasis [12,22]. In our present study, the results are not statistically significant due to the small number of cases included in the study. Continued studies are needed for further investigation. Furthermore, the emergences of fluconazole-resistant *C. albicans* and echinocandin-resistant strains due to the FKS gene have been reported [26,27]. As mentioned above, however, no strains suspected of being highly drug-resistant were observed at our hospital. That result suggests that drug resistance has not contributed to the complications of ocular lesions at our institution. Monitoring the breakdown of the candida species isolated at each facility and antifungal drug susceptibility would be important from a therapeutic standpoint, such as in the selection of antifungal drugs for the overall treatment of candidemia.

The majority of candidemia is caused by catheter-related bloodstream infection (CRBSI), and indwelling CV catheters are one of the most important risk factors [3]. Therefore, the prompt removal of catheters is strongly recommended in cases of candidemia. In this study, we showed that not removing catheters increased the risk for developing ocular candidiasis. Candida forms biofilms on catheters, and it is known that in biofilms, antifungal drugs do not reach concentrations sufficient to exert an antibacterial effect [28,29,30]. It is notable that a higher percentage of CV ports were found in catheters that were not removed. Generally, CV ports tend not to be removed immediately after the onset of CRBSI by comparison with CVC because this requires a surgical procedure. Early removal of catheters in patients with candidemia should be recommended because it not only improves prognosis but also may contribute to a lower incidence of ocular candidiasis.

βDG is an adjunct diagnosis for fungal infections and is sometimes used to determine treatment efficacy [31]. The relationship between a high βDG value and ocular candidiasis has been reported, but few reports have shown an association between the time from the onset of candidemia to the diagnosis of ocular candidiasis and βDG levels [12]. In the present study, we clarified the occurrence of high βDG levels as an independent risk factor for ocular candidiasis. The timing of ophthalmology consultations was not standardized in this study, which is a limitation, but use of the Kaplan−Meier curve showed that the proportion of cases diagnosed with ocular candidiasis increased over time in the group with higher values for βDG. This result indicates that repeat ophthalmologic examinations may help with the diagnosis of ocular candidiasis in cases of candidemia with high values for βDG.

Though complications of ocular candidiasis were not extracted as a statistically independent risk factor for 30-day mortality, patients with ocular candidiasis experienced a higher mortality rate compared with uncomplicated cases. Some of the patients who were excluded from the study because of no ophthalmologic examination included early death cases in the course of the disease, so the correlation between ocular involvement and prognosis may not have been fully evaluated. In fact, the 30-day mortality rate in the 138 cases was 32.3%, which included patients with no ophthalmologic examination. On the other hand, the 30-day mortality rate for the 108 cases that were ultimately included in the study was 20.4%. There were some reports that the mortality rate was higher in cases with ocular lesions, and although the ocular lesions themselves may not affect the prognosis, the intensity of the disease caused by disseminated lesions is expected to have an effect on the prognosis. Careful management of patients with ocular candidiasis is desirable due to the risk of deterioration in their general condition.

We were unable to extract any factors that would make ocular candidiasis less likely to occur in this analysis. Therefore, we recommend that ophthalmological examination be performed in all cases presenting with candidemia. Even after a negative blood culture is confirmed, ophthalmological examination should be repeated in cases of candidemia and ocular candidiasis, and if new lesions have appeared or the lesions do not improve, the antifungal drug should be changed based on tissue migration and drug sensitivity, and treatment should be continued until the fundus findings improve.

## 5. Conclusions

Repeated searches for ocular lesions by ophthalmological examination is an essential strategy in cases of candidiasis, and we recommend the first session should occur within 7 days of onset, while a second session should be scheduled for 7–14 days following onset. Furthermore, it is also necessary to constantly check for the presence of subjective eye symptoms. Particular attention should be paid to complications of ocular lesions in cases with the following factors: isolation of *C. albicans*, unremoved CVC, and a high level of βDG. Since the prognosis may be poor in cases with ocular involvement, we also recommend careful systemic management of those patients.

## Figures and Tables

**Figure 1 jof-08-00497-f001:**
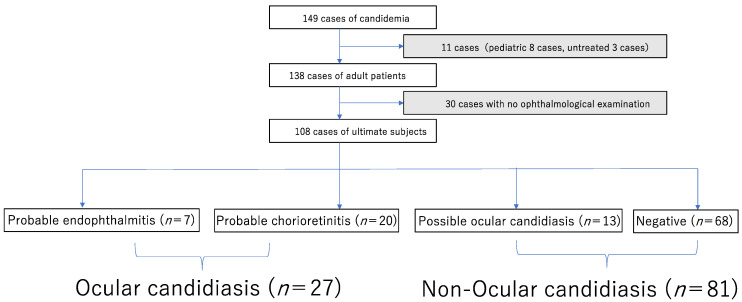
The process of selecting and classification target cases. Gray shadow: excluded cases. The ultimate analysis included 108 cases. Twenty-seven cases of ocular candidiasis included seven probable endophthalmitis and twenty probable chorioretinitis. Eighty-one cases of non-ocular candidiasis included thirteen possible ocular candidiasis and sixty-eight cases with no sign of ocular candidiasis.

**Figure 2 jof-08-00497-f002:**
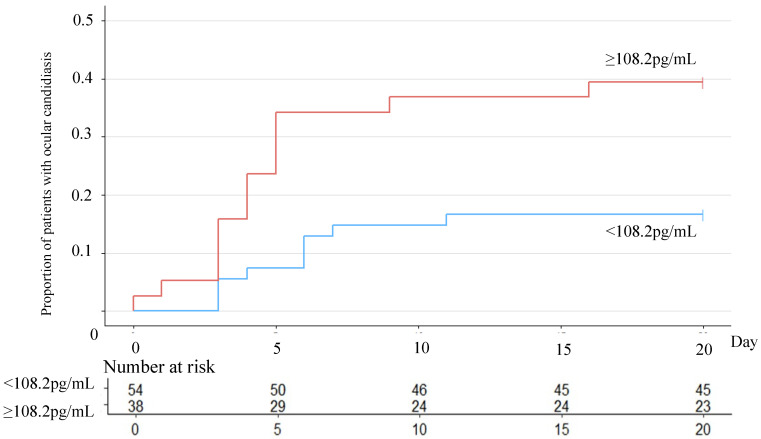
Kaplan−Meier curve showing the proportion of patients with ocular candidiasis in two groups determined by β-D-glucan levels. ROC curves were created for the presence of ocular lesions, and the Youden index was calculated to be 108.2. Based on this β-D-glucan value, cases were divided into two groups. The log-run test was statically significant at *p* < 0.05.

**Table 1 jof-08-00497-t001:** Characteristics of patients with ocular candidiasis and the intervals of ophthalmologic examinations.

Age	Sex	*Candida* spp.	Eye Symptom	Onset to 1st Exam (Days)	1st Exam to 2nd (Days)	Diagnosis at 2nd Exam	Background
**Probable Endophthalmitis**							
73	F	*Candida* sp.	Poor eyesight	0	3	No	Thyroid cancer
79	F	*C. albicans*	Poor eyesight	12	3	No	Stomach cancer
66	M	*C. krusei*	Poor eyesight	2	7	**Yes**	AML, severe neutropenia
76	M	*C. albicans*	Myodesopsia	5	9	No	Pharyngeal cancer
69	M	*C. albicans*	Poor eyesight	4	4	**Yes**	Pancreatic cancer
37	M	*C. albicans* *C. famata*	Poor eyesight Misty vision	1	Died	No	HIV infection
64	M	*C. albicans*	Poor eyesight Myodesopsia	3	7	No	Stomach cancer
**Probable Chorioretinitis**							
65	F	*C. famata*	No	5	10	No	Peritoneal cancer
74	M	*C. albicans*	No	11	12	No	Bile duct cancer
57	F	*C. albicans*	No	3	9	**Yes**	SAH
75	M	*C. parapsilosis* *C. glabrata*	No	6	Died	No	Multiple trauma
56	F	*C. albicans*	No	3	8	No	Acute aortic dissection
77	F	*C. albicans*	No	4	20	No	Rheumatoid arthritis
46	M	*C. krusei*	No	16	Died	No	Burkitt’s lymphoma
71	M	*C. parapsilosis*	No	6	2	No	Cervical abscess
73	F	*C. albicans*	No	4	Died	No	Mandibular cancer
63	F	*C. albicans*	No	3	6	No	Pulmonary embolism
66	M	*C. parapsilosis*	No	6	5	No	SAH
73	M	*C. albicans*	No	5	8	No	Ischemic heart disease
78	M	*C. albicans* *C. glabrata*	No	3	8	No	Pancreas cystic tumor
80	M	*C. albicans*	No	4	3	No	Infectious endocarditis
69	F	*C. glabrata*	No	5	7	No	Cervical cancer
57	F	*C. albicans*	No	3	Died	No	Ovarian cancer
78	M	*C. albicans*	No	5	14	No	Esophageal cancer
81	M	*C. albicans*	No	4	Died	No	Malignant melanoma
43	M	*C. albicans*	No	3	7	No	Acidophilic enteritis
68	M	*C. albicans*	No	7	5	No	Colon cancer

AML, acute myeloid leukemia; HIV, human immunodeficiency virus; SAH, Sub-arachnoid hemorrhage.

**Table 2 jof-08-00497-t002:** Univariate analysis for the factors with ocular candidiasis in cases with candidemia.

	Ocular Candidiasis (*n* = 27)	(%)	Non-Ocular Candidiasis (*n* = 81)	(%)	*p*-Value
Mean age ± SD	67.1 ± 11.5		68.1 ± 12.8		0.63
Male	17	62.9%	54	66.6%	0.72
Underlying diseases					
Diabetes mellitus	5	18.5%	31	38.2%	0.06
Hypertension	10	37.0%	36	44.4%	0.50
Chronic heart disease	5	18.5%	16	19.7%	1.00
Chronic kidney disease stage**Ⅴ**	2	7.4%	9	11.1%	0.72
Liver diseases	1	3.7%	10	12.3%	0.28
COPD	1	3.7%	3	3.7%	1.00
Malignancy	15	55.5%	41	50.6%	0.65
Hematological malignancy	2	7.4%	12	14.8%	0.51
Solid-organ cancer	13	48.1%	29	35.8%	0.25
Immunocompromised status					
Steroid therapy	4	14.8%	15	18.5%	0.77
Immunosuppressive agent	1	3.7%	7	8.6%	0.67
Chemotherapy	9	33.3%	23	28.4%	0.62
Neutropenia	2	7.4%	7	8.6%	1.00
Radiation therapy	0	0.0%	8	9.8%	0.19
HIV infection	1	3.7%	0	0.0%	0.25
Stem cell transplantation	1	3.7%	5	6.1%	1.00
Microbiology					
*C. albicans*	18	66.7%	28	34.6%	<0.05
*C. parapsilosis*	3	11.1%	24	29.6%	0.07
*C. glabrata*	3	4.0%	13	15.0%	0.75
*C. krusei*	2	8.0%	3	3.7%	0.59
*C. tropicalis*	0	0.0%	5	6.5%	0.33
*C. famata*	2	8.0%	4	5.0%	0.62
Other *Candida species*	1	3.7%	5	6.1%	1.00
Polymicrobial fungemia	2	7.4%	1	1.2%	0.15
Underlying conditions					
ICU admission	11	40.7%	31	38.2%	0.81
Surgery	7	25.9%	29	35.8%	0.34
Abdominal surgery	2	7.4%	16	19.7%	0.23
Open-heart surgery	2	7.4%	6	7.5%	1.00
Prior antibiotics exposure	24	88.8%	59	28.8%	0.08
Prior antifungal exposure	2	7.4%	14	17.2%	0.34
Indwelling CVC	23	85.1%	60	74.0%	0.23
unremovable of CVC	5	18.5%	3	3.7%	<0.05
nterval days onset and removal of CVC	1.1 ± 1.6		2.1 ± 2.6		
TPN	17	62.9%	48	59.2%	0.73
Mechanical ventilation	5	18.5%	18	22.2%	0.79
Hemodialysis	4	14.8	14	17.2	1.00
Septic shock	6	22.2%	17	20.9%	0.89
Initial antifungal drugs					
Azole	15	55.6%	49	60.5%	0.65
Echinocandin	9	33.3%	31	38.3%	0.64
L-AMB	2	7.4%	1	1.2%	0.15
Interval days onset and administering antifungal drugs	2.8 ± 0.38		2.2 ± 0.22		0.33
Persistent BSI	11	44.0%	27	33.8%	0.36
(1,3)-β-D-glucan					<0.05
30-day mortality	10	37.0%	12	14.8%	<0.05

SD, standard deviation; COPD, chronic obstructive pulmonary disease; HIV, human immunodeficiency virus; ICU; intensive care unit; CVC, central venous catheter; TPN, total parenteral nutrition; BSI, blood stream infection; L-AMB, liposomal amphotericin B.

**Table 3 jof-08-00497-t003:** Multivariate analysis for the factors with ocular candidiasis in cases with candidemia.

	OR	95% CI	*p*-Value
*C. albicans*	4.85	1.58–14.90	<0.05
Unremoved CVC	10.40	1.74–62.16	<0.05
(1,3)-β-D-G-glucan	1.003	1.0004-1.005	<0.05

OR, odds ratio; CI, confidence interval; CVC, central venous catheter.

**Table 4 jof-08-00497-t004:** Univariate analysis of factors associated with 30-day mortality.

	Survivors (*n* = 86)	(%)	Death (*n* = 22)	(%)	*p*-Value
Mean age ± SD	68.1 ± 12.3		67.1 ± 13.7		0.75
Male	61	70.9%	10	45.5%	<0.05
Underlying diseases					
Diabetes mellitus	29	33.7%	7	31.8%	0.86
Hypertension	38	44.2%	8	36.4%	0.50
Chronic heart disease	17	19.8%	4	18.2%	0.86
Chronic kidney disease stage**Ⅴ**	10	11.6%	1	4.6%	0.28
Liver diseases	11	12.7%	0	0.0%	0.11
COPD	2	2.3%	2	9.1%	0.18
Malignancy	44	51.2%	12	54.6%	0.77
Hematological malignancy	10	11.6%	4	18.1%	0.43
Solid-organ cancer	34	39.5%	8	36.4%	0.78
Immunocompromised status					
Steroid therapy	13	15.1%	6	27.3%	0.20
Immunosuppressive agent	5	5.8%	3	13.6%	0.35
Chemotherapy	24	27.9%	8	36.3%	0.44
Neutropenia	6	7.0%	3	13.6%	0.38
Radiation therapy	8	9.3%	0	0.0%	0.20
HIV infection	0	0.0%	1	4.6%	0.20
Stem cell transplantation	4	4.7%	2	9.1%	0.59
Microbiology					
*C. albicans*	36	41.9%	10	45.5%	0.81
*C. parapsilosis*	22	25.6%	5	22.7%	0.78
*C. glabrata*	13	15.1%	3	13.6%	0.86
*C. krusei*	3	3.5%	2	9.1%	0.30
*C. tropicalis*	4	4.7%	1	4.6%	0.98
*C. famata*	4	4.7%	2	9.1%	0.41
Other *Candida species*	6	7.0%	0	0.0%	0.34
Polymicrobial fungemia	2	2.3%	1	4.6%	0.49
Underlying conditions					
ICU admission	35	40.7%	7	31.8%	0.47
Surgery	31	36.1%	5	22.7%	0.22
Abdominal surgery	14	16.3%	4	18.2%	0.83
Open-heart surgery	8	9.41%	0	0.0%	0.20
Prior antibiotics exposure	65	75.6%	18	81.8%	0.52
Prior antifungal exposure	12	14.0%	4	18.2%	0.62
Indwelling CVC	67	77.9%	16	72.7%	0.61
Unremovable of CVC	2	2.3%	6	27.3%	<0.05
Interval days onset and removal of CVC	1.8 ± 0.3		2.5 ± 0.8		0.39
TPN	51	59.3%	14	63.6%	0.70
Mechanical ventilation	19	22.1%	4	18.2%	0.68
Hemodialysis	10	11.6	1	4.6%	0.28
Septic shock	16	14.8%	7	6.5%	0.19
Initial antifungal drugs					
Azole	11	50.0%	53	61.6%	0.32
Echinocandin	10	45.5%	30	34.9%	0.36
L-AMB	1	4.6%	2	2.3%	0.59
Interval days onset and administering antifungal drugs	2.4 ± 0.21		1.8 ± 0.42		0.19
Persistent BSI	13	65%	25	29.4%	<0.05
(1,3)-β-D-glucan					<0.05
Ocular candidiasis	17	19.8%	10	45.5%	<0.05

SD, standard deviation; COPD, chronic obstructive pulmonary disease; HIV, human immunodeficiency virus; ICU, intensive care unit; CVC, central venous catheter; TPN, total parenteral nutrition; BSI, blood stream infection; L-AMB, liposomal amphotericin B.

**Table 5 jof-08-00497-t005:** Multivariate analysis of the factors associated with 30-day mortality.

	OR	95% CI	*p*-Value
Female	1.75	0.52–5.81	0.35
Unremoved CVC	17.76	2.18–144.38	<0.05
Persistent BSI	3.17	0.79–12.70	0.10
(1,3)-β-D-G-glucan	1.00	0.90–1.00	0.22
Ocular candidiasis	1.98	0.54–7.34	0.30

R, odds ratio; CI, confidence interval; CVC, central venous catheter; BSI, blood stream infection.

## Data Availability

Not applicable.

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
