# Peer review of "Risk Factors and Clinical Characteristics of Patients with Ocular Candidiasis"

_jof, 2022, doi:10.3390/jof8050497_

Round 1

Reviewer 1 Report

It is an interesting study that complements other studies carried out on the same subject.

It is necessary to aggregate the results of the univariate and multivariate analyzes in tabular form in order to discuss the results of similar studies.

Was the risk of mortality at 30 days associated with the species involved?, or perhaps with the presence of ocular candidiasis?, or a combination of several factors? What results does the multivariate analysis yield?

Some cases of candidemia can have a fatal prognosis in the first and second week of hospital admission. Are there variations in the univariate and multivariate analysis in relation to the time of death (for example, in one, 2 or 4 weeks)?

What treatments did the ocular cases, deceased and non-deceased, receive?

Reviewer 2 Report

This is a very valuable study providing important data on ocular involvement of candidiasis.

Major comments

Was duration of blood culture positivity collected? It has been postulated that complications are more likely to occur in patients with prolonged blood culture positivity compared with patients who have eg a single positive blood culture and then clear the candidaemia. This data would be valuable to include.

Please include numbers of patients with other complications of candidaemia - in particular endocarditis. This is important to place the current patient cohort into the context of what is described in the literature.

Please include a description of initial treatment. Were patients treated with an echinocandin more likely to develop vitro tissue than those treated with an azole?

How many days did patients have eye symptoms  prior to assessment confirming vitiritis?

Was candidaemia source evaluated? It is likely that patients with a urinary source are less likely to develop ocular candidiasis and these patients probably don’t warrant eye assessment. 

The discussion comments on the higher likelihood of c albicans in ocular candidiasis. The numbers for Candida species other than albicans and parapsilosis are small limiting further conclusion. Note must be made of this and appropriate adjustments to the discussion. Eg has c tropicalis previously been associated with ocular candidiasis ? 

Please also include in the discussion comment on which patients might be safe to not have an eye examination eg urinary source, rapidly clear blood cultures, low BDG.

Is chorioretinitis worthwhile identifying? Please include a comment in the discussion and expand on how this would change management or monitoring.

Minor comments

Methods Line 121

What were the criteria for inclusion variables in the multivariate analysis? More detail needed “regression multivariate analysis with odds ratios (OR) and 95% confidence intervals (CI).”

Line 163 of 3.2. Clinical characteristics of ocular candidiasis

“Seven” does not need to be capitalised

Line 182 

As far as I can tell there was one patient with AML and neutropaenia. Make this singular - “in patients with AML and severe neutropenia”

Table 3 has a spelling error - Kurusei

Round 2

Reviewer 1 Report

Thank you for responding in such a short time.
I believe that the corrections made fill the gaps that I observed in the study.